# 3DCoMPaT200: Language-Grounded Compositional Understanding of Parts and Materials of 3D Shapes

**Mahmoud Ahmed**[1]     **Xiang Li**[1]     **Arpit Prajapati**[2]     **Mohamed Elhoseiny**[1]

[1]KAUST (King Abdullah University of Science and Technology)
[2]Poly9

{mahmoud.ahmed, xiang.li.1, mohamed.elhoseiny}@kaust.edu.sa
arpit@polynine.com

## Abstract

Understanding objects in 3D at the part level is essential for humans and robots to navigate and interact with the environment. Current datasets for part-level 3D object understanding encompass a limited range of categories. For instance, the ShapeNet-Part and PartNet datasets only include 16, and 24 object categories respectively. The 3DCoMPaT dataset, specifically designed for compositional understanding of parts and materials, contains only 42 object categories. To foster richer and fine-grained part-level 3D understanding, we introduce 3DCoMPaT200, a large-scale dataset tailored for compositional understanding of object parts and materials, with 200 object categories with $\approx$5 times larger object vocabulary compared to 3DCoMPaT and $\approx$ 4 times larger part categories. Concretely, 3DCoMPaT200 significantly expands upon 3DCoMPaT, featuring 1,031 fine-grained part categories and 293 distinct material classes for compositional application to 3D object parts. Additionally, to address the complexities of compositional 3D modeling, we propose a novel task of Compositional Part Shape Retrieval using ULIP to provide a strong 3D foundational model for 3D Compositional Understanding. This method evaluates the model shape retrieval performance given one, three, or six parts described in text format. These results show that the model's performance improves with an increasing number of style compositions, highlighting the critical role of the compositional dataset. Such results underscore the dataset's effectiveness in enhancing models' capability to understand complex 3D shapes from a compositional perspective. Code and Data can be found here: https://github.com/3DCoMPaT200/3DCoMPaT200/

## 1 Introduction

Part-level 3D object understanding is a fundamental aspect of machine perception, enabling systems to interpret and interact with the three-dimensional world in a manner akin to human cognition. Humans intuitively recognize, categorize, and manipulate objects based on their constituent parts and the materials from which they are made. This level of understanding underpins everyday actions such as opening doors or using tools, which rely on recognizing parts (e.g., doorknobs, handles) and their properties. In computational terms, imparting machines with similar capabilities is essential for a host of applications in computer vision, robotics, and interactive simulation. Despite its importance, part-level 3D understanding remains a challenging domain due to the complexity of objects and the intricacies of their materials and shapes.

Recent advancements have seen the development of comprehensive datasets aimed at fostering 3D visual understanding. Notably, datasets like ShapeNet [1], ModelNet [2], have provided valuable resources for analyzing the geometric shapes of objects. Subsequently, 3D-FUTURE [3] introduced a collection focused on industrial 3D CAD shapes of furniture, enriched with textures designed by

38th Conference on Neural Information Processing Systems (NeurIPS 2024) Track on Datasets and Benchmarks.

professionals. Objaverse [4] and Objaverse-XL [5] increase the scale of 3D objects with more than 10 million 3D objects. Despite these notable efforts to advance 3D understanding, recent object-centric 3D datasets lack part-level annotations. ShapeNet-Part [6] extends ShapeNet [1] by introducing part-level annotations; however, it offers only coarse-level part annotations. On the other hand, PartNet [7] enhances ShapeNet [1] by offering fine-grained labels for part segmentation, yet it too lacks information regarding the materials. Material properties are not only essential for rendering realistic images but also provide crucial semantic information that influences the functionality and categorization of objects. The nuanced understanding of materials, combined with geometric shapes, can dramatically enhance the realism and applicability of simulated environments, contributing to more effective training and deployment of machine learning models.

Addressing this gap, the 3DCoMPaT [8, 9] dataset represented a notable step by integrating part-level material information, offering an unprecedented array of rendered model styles and material classifications. This dataset opened new avenues for research and application, particularly in enhancing the realism of 3D renderings and supporting more accurate simulations. However, despite its advancements, 3DCoMPaT [8, 9] is constrained by its limited scope in object categories, which narrows its utility and applicability in addressing the full spectrum of real-world scenarios.

To address these challenges, we present 3DCoMPaT200, an extensively curated dataset that builds upon 3DCoMPaT [8, 9], incorporating 19k 3D shapes across 200 diverse object categories. This dataset is enriched with 1,031 intricate part categories and 293 unique material classes, enabling compositional part-material understanding of 3D objects. 3DCoMPaT200 substantially broadens the range, introducing a significantly greater number of object categories and parts in both fine and coarse-grained semantics as shown in Fig. 2 For a comprehensive comparison between 3DCoMPaT200 and earlier datasets, refer to Table 1. Details for the data collection, rendering pipelines and data examples are provided in the appendix.

To study the importance of the Compositions in the dataset, we conduct experiments for multi-modality alignment using ULIP [10, 11] with captions generated from our compositional metadata information covering the shape name, part names, style of each part, and the associated color. We evaluate multi-modal alignment between text and 3D using ULIP on different numbers of parts per caption on one composition to measure the level of model compositional understanding in retrieving the correct shape given the detailed caption. Our contributions are summarized as follows:

- We present 3DCoMPaT200, a novel dataset designed for compositional 3D understanding of object parts and materials. This dataset encompasses 200 object categories, representing 5 orders of magnitude increase and 1,031 fine-grained parts accounting for 4 times the number of parts compared to its predecessor, 3DCoMPaT.

- We conduct extensive evaluations to benchmark leading methodologies in the realms of object classification, part-material segmentation, and Grounded Compositional Recognition (GCR) on the 3DCoMPaT200 dataset. These assessments underscore the dataset's capacity to propel the development of more nuanced and effective 3D object understanding techniques.

- We propose a novel Compositional Shape Retrieval task that aims to retrieve 3D shapes from compositional part-material descriptions and benchmark the current state-of-the-art method.

## 2 Related Work

### 2.1 Part-level 3D Object Understanding Datasets

Recent developments in 3D visual understanding have produced several influential datasets like ShapeNet [1], ModelNet [2], and Objaverse [4], which provide substantial resources for analyzing object geometries. The 3D-FUTURE dataset [3] offers over 10,000 textured CAD models of furniture designed by professionals, enhancing the connection between virtual and real-world applications. Newer resources such as OmniObject3D [15] and ABO [13] offer high-quality 3D assets with comprehensive annotations for realistic 3D perception and generation tasks. The Objaverse series, particularly Objaverse-XL [5], expands the scale dramatically with more than 10 million 3D objects, further enriching the landscape of available 3D data.

| Dataset | Shapes | | | Materials | | Parts | | | Images | |
|---|---|---|---|---|---|---|---|---|---|---|
| | Count | Stylized | Classes | Count | Classes | Instances/Shape | Hierarchical | Instances | Count | Alignment |
| ModelNet [2] | 128K | ✗ | 662 | ✗ | ✗ | ✗ | ✗ | ✗ | ✗ | ✗ |
| ShapeNet-Core [1] | 51,3K | ✗ | 55 | ? | ? | ✗ | ✗ | ✗ | ✗ | ✗ |
| ShapeNet-Sem [1] | 12K | ✗ | 270 | ? | ? | ✗ | ✗ | ✗ | ✗ | ✗ |
| ObjectNet3D [12] | 43,2K | ✗ | 100 | ✗ | ✗ | ✗ | ✗ | ✗ | 90K | pseudo |
| 3D-Future [3] | 9,9K | ✗ | 15 | ? | 15 | ✗ | ✗ | ✗ | 20K | exact |
| ABO [13] | 148K | ✗ | 98 | ✗ | ✗ | ✗ | ✗ | ✗ | 398K | pseudo |
| Objaverse-XL [5] | 10,2M | ✗ | ? | ? | ? | ✗ | ✗ | ✗ | 66M | exact |
| ShapeNet-Part [6] | 31,9K | ✗ | 16 | ✗ | ✗ | 2.99 | ✗ | ✗ | ✗ | ✗ |
| ShapeNet-Mats [14] | 3,2K | ✗ | 3 | ? | 6 | 6.2 | ✗ | ✗ | ✗ | ✗ |
| PartNet [7] | 26,7K | ✗ | 24 | ✗ | ✗ | 18 | ✓ | ✓ | ✗ | ✗ |
| 3DCoMPaT [8] | 7,2K | 7,2M | 42 | 167 | 11 | 5.12 | ✗ | ✗ | 58M | exact |
| 3DCoMPaT++ [9] | 10K | 10M | 41 | 293 | 13 | 9.98 | ✓ | ✓ | 160M | exact |
| **3DCoMPaT200** | 19k | 19M | 200 | 293 | 13 | 5.16 | ✓ | ✓ | 304M | exact |

Table 1: Comparison of 3DCoMPaT200 with existing commonly used 3D object understanding datasets. We show datasets without part annotations at the top and datasets with part annotations at the bottom.

Despite these advances, a persistent shortfall remains in the provision of detailed material information at the part level. While ShapeNet-Part [6] introduced part-level annotations, it lacks material details. PartNet [7] offers fine-grained part segmentations but also omits material information. The 3DCoMPaT datasets [8, 9] address these limitations by integrating comprehensive material details into the part-level annotations and offering a rich array of rendered styles and material categories, yet they cover a relatively limited range of object categories.

In response, our newly proposed 3DCoMPaT200 dataset significantly expands the scope to include 200 object categories, providing high-quality annotations that combine detailed material properties with geometric shapes. This expansion is crucial for enhancing the realism and applicability of 3D simulations, thereby improving both training efficiency and the deployment effectiveness of machine learning models in real-world scenarios.

## 2.2 Grounded Compositional Grounding

In order to capture a well-rounded evaluation of models' ability in 3D compositional understanding, 3DCoMPaT++[9] presented a set of new metrics for Grounded Compositional Recognitions(GCR). The GCR task leverages compositional metrics adapted from those used in image-based activity recognition [16, 17], emphasizing both the accuracy of part-material pair predictions and the precision of their segmentation. These metrics include Shape Accuracy for correct shape categorization, Value for accurate part-material pair predictions, Value-All for the accuracy of predicting all part-material pairs of a shape correctly, and their grounded variants—Grounded-Value and Grounded-Value-All—where the part is correctly segmented. Specifically, segmentation quality is measured by the intersection over union (IoU) with ground truth. The detailed evaluation of these metrics ensures that both the classification and the segmentation of parts and materials are thoroughly assessed, providing a robust framework for analyzing complex 3D models. In our experiments, we aim to compute these metrics on the larger collection of shapes but we mainly depend on the 3D colored point clouds without using a multimodal model like BPNet [18].

## 2.3 Text-3D Retrieval

Recent works have demonstrated the effectiveness of extending multimodal representation learning into the 3D domain. Methods such as ULIP2 [10] integrate 3D point clouds with image and language modalities, enhancing 3D representation learning and reducing reliance on densely annotated 3D data. The model relies on training on trimodal datasets from ShapeNet [1] and Objaverse [4, 5]. However, the captions used were generated from BLIP2 [19] which often lacks a detailed compositional description of the 3D shape. Another work [20] builds on ULIP's [11] methodology to enhance performance, though they depend on manual 3D data annotations and a complex data engineering

process. Similar works have been pushing the alignment performance on 3D shapes using full ViT [21] model for the backbone of the 3D encoder [22] and using multiview rendered images to represent the 3D shape [23]. A persistent challenge in this field is the difficulty of compiling scalable, high-quality, and well-aligned multimodal data sets for 3D applications. To address this issue, we leverage our heavily annotated dataset metadata to compile a captioning dataset for each shape with a scalable feature that accommodates for the number of compositions. We also, present a new benchmark for Compositional Shape Retrieval measuring the retrieval scores depending on the number of compositions used for evaluation.

## 3 3DCoMPaT200

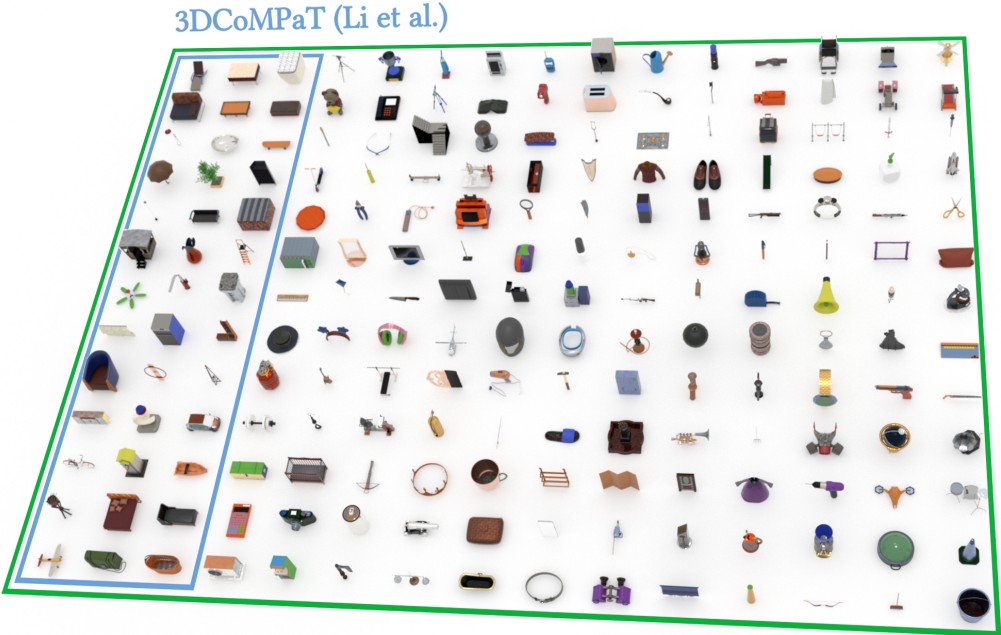

Figure 1: Illustrating the 3DCoMPaT200 expansion over 3DCoMPaT [8, 9]. Each rendered shape represents a single shape class. Our iteration of the dataset contains nearly 160 additional shape classes and 10 thousand additional shapes, all meticulously annotated at the part-instance level.

### 3.1 Data Summary

The 3DCoMPaT200 dataset is a comprehensively annotated collection of 3D objects, featuring artist-designed CAD models equipped with high-quality, part-level annotations. The dataset comprises 19k 3D shapes from 200 object categories, each annotated with both fine-grained and coarse-grained part labels. We have 118 coarse-level part labels and 1031 fine-level part labels. Human annotators have meticulously gathered compatible materials for each designated part, facilitating the creation of styled shapes. In our dataset, we have 13 coarse-grained materials and 293 fine-grained materials. For every shape, we further supply 8 2D rendered images captured from both standard and random viewpoints. Accompanying each rendered image, we also offer corresponding depth maps, part maps, and material maps. All annotations have been conducted by professionally trained annotators, adhering to a stringent, multi-stage review protocol. Figure 1 shows examples of styled shapes from our 3DCoMPaT200 dataset.

### 3.2   Data Collection

Our 3DCoMPaT200 dataset is collected following the same pipeline as the original 3DCoMPaT dataset while carefully extending instructions to an additional 158 categories, including their part definitions and examples. Across these categories, we maintain an average number of shapes of 60 with an average number of fine-grained parts of 5.12 per shape. The dataset's compilation process entails a detailed and structured approach, beginning with the collection and refinement of CAD shapes by a collaborating industry partner to ensure uniformity and quality. This is followed by detailed part annotations to each 3D model, adhering to category-specific guidelines to ensure accuracy and consistency. Materials are then meticulously assigned to each shape part from a predefined set of classes, ensuring compatibility and coherence. Subsequently, each shape is stylized by randomly assigning materials to create various styles, enhancing the dataset's diversity. Each set of material-shape assignments makes up a single composition and using the fine-grained materials we create up to 1000 compositions for our dataset. Finally, shapes are rendered from multiple perspectives, accompanied by corresponding masks, depth maps, and point cloud data, to create a comprehensive and versatile dataset. To mitigate the high granuality of the fine-grained part labels, we further annotated them into coarse-grained parts. The primary distinction lies in their level of detail: fine-grained parts contain multiple shape-specific elements suitable for tasks requiring detailed, category-level comprehension, while coarse-grained parts are more generalized to support tasks needing higher-level shape understanding. This is evident in the decrease in category-specific part representation, from 84.19% in fine-grained semantics to 35.59% in coarse-grained semantics.

**Data Collection Efforts**. The preparation of annotation guidelines for 200 object categories by 3D experts requires approximately 1 hour per category. For geometry correction, part and material annotation, each model demands roughly 2 hours of human effort. Overall, the total time invested in collecting annotations for the entire dataset amounts to approximately 38,600 hours. For 2D image rendering, it takes around 1,380 CPU hours to render all 304M images.

### 3.3   Long-tail Distribution

In our 3DCoMPaT200 dataset, we observe a significant long-tail distribution of object categories and part occurrences, mirroring real-world frequency and variety. This dataset encompasses a wide array of commonly recognized object categories, leading to a diverse range of shapes and structures. However, similar to challenges faced in other extensive datasets, such as ScanNet200, the distribution of these categories and their corresponding parts is not uniform.

Figure 2 illustrates the class distribution, highlighting the variance in part occurrences and object category frequencies. Significant disparities exist in the occurrence rates of different object categories and parts. For instance, common objects like chairs and tables appear more frequently than less common items like sword, camera, and fishing rod. This imbalance extends to the part-level annotations, where certain common parts (e.g., legs, tops) are more prevalent than unique, object-specific parts (e.g., piano keys, tool handles). This visualization underscores the necessity of adopting specialized approaches to effectively leverage the rich annotations and diversity present in the 3DCoMPaT200 dataset, enabling advanced research and development in 3D object understanding.

## 4   Experiments and Results

In this section, we conduct experiments on our 3DCoMPaT200 dataset on multiple object understanding tasks, including shape classification, part segmentation, material segmentation, grounded compositional recognition tasks, and Text Part Shape Retrieval. The following results are done on one set of random assignment of material-part suitable pairs which we refer to as a composition. The one composition of the dataset currently accounts for 19,051 shapes. The mentioned shapes are divided into 3 splits: training, validation, and testing each with 14,350, 1,692, and 3,009 shapes respectively. All results are reported on a testing set of one composition and models were trained on one composition as well. .

### 4.1   3D Shape Classification

As depicted in Figure 2, our dataset presents a long-tail distribution. We perform experiments in 3D shape classification and part segmentation using the XYZ coordinates of the shapes. The point clouds,

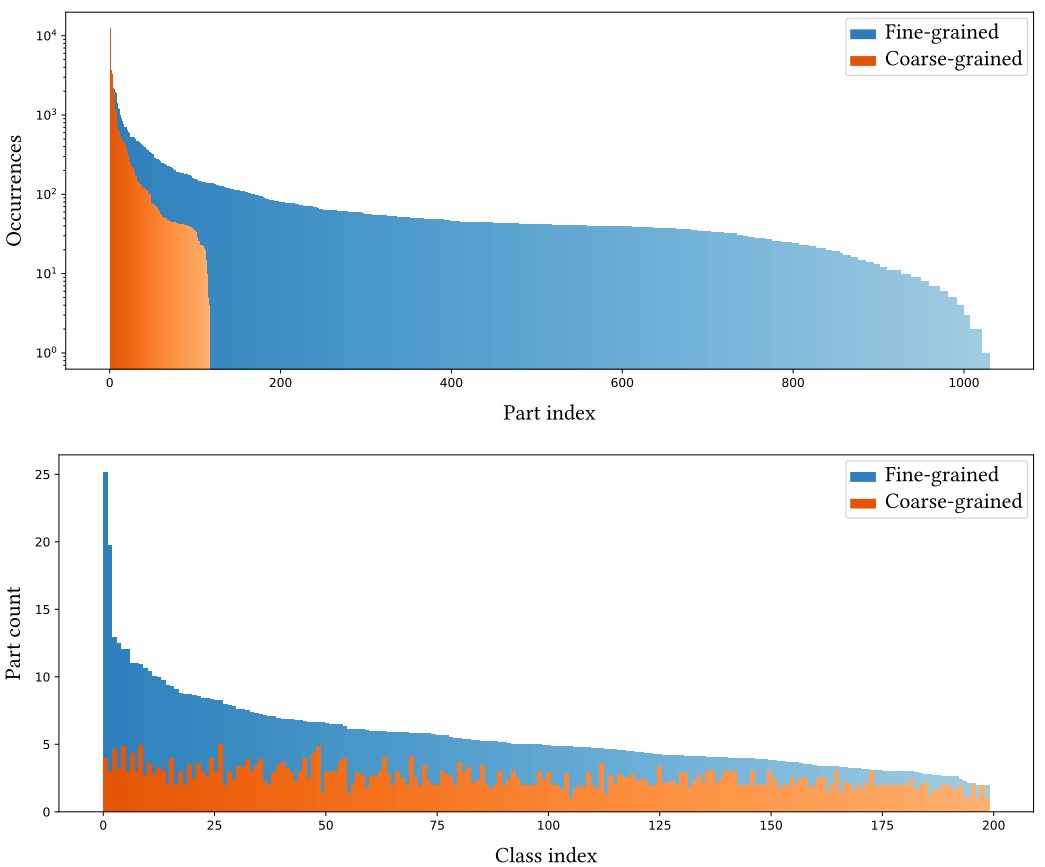

Figure 2: **Top:** We plot the distribution of part occurrences at both fine (blue) and coarse (red) granularity levels, in *log scale*. **Bottom:** We plot the distribution of part occurrences across all part categories.

sampled at a size of 2048 from the meshes, serve as the primary data format for these experiments. In shape classification, as shown in Table 2, models such as PCT, PointNet++, CurveNet, PointBert, PointTransformer V2, and PointTransformer V3 are evaluated. Despite the skewed data distribution, PointTransformer V3 and PointNet++ achieve the highest accuracies, with PointTransformer V3 obtaining a top accuracy of 88.48% and PointNet++ following closely with 88.32%.

Table 2: Comparison of model performances in terms of instance accuracy and class average accuracy.

| Model | Instance Acc (%) | Class Avg Acc (%) |
|---|---|---|
| PointNet++[24] | **88.32** | **89.02** |
| PCT[25] | 71.71 | 71.50 |
| CurveNet[26] | 86.00 | 86.77 |
| PointBert[27] | 87.65 | 87.84 |
| PointTransformer V2[28] | 85.95 | 86.35 |
| PointTransformer V3[29] | **88.48** | **89.79** |

## 4.2 Part and Material Segmentation

For part segmentation, we adopt the same models to distinguish between coarse and fine-grained segments. Introducing an embedding of class labels for each shape—referred to as a *shape prior*—significantly simplifies the task in fine-grained segmentation. This adaptation results in a noticeable improvement in segmentation performance, as illustrated in Table 3, with PCT reaching a maximum mean Intersection over Union (mIoU) of 81.83% with the shape prior and 48.70% without

it. The use of the shape prior effectively conditions the model on specific part classes, enhancing segmentation performance across all models.

Table 3: Models performance on Part segmentation evaluated on 10 Compositions where we report the Accuracy and mIoU. Prior stands for the usage of Shape Class Embedding in the model

| Model | Shape Prior | Fine Grained | | Coarse Grained | |
|---|---|---|---|---|---|
| | | Acc. (%) | mIoU (%) | Acc. (%) | mIoU (%) |
| PointNet++ [24] | ✓ | 80.39 | 54.34 | 92.3 | 77.55 |
| PointNet++ [24] | ✗ | 70.30 | 45.44 | 65.25 | 43.35 |
| PCT [25] | ✓ | 86.21 | **81.83** | **96.42** | **91.83** |
| PCT [25] | ✗ | 62.46 | 48.70 | 84.74 | 76.61 |
| Curvenet [26] | ✓ | 83.07 | 59.69 | 93.64 | 80.42 |
| Curvenet [26] | ✗ | 71.36 | 50.97 | 88.08 | 72.74 |
| PointBert [27] | ✓ | 81.07 | 62.93 | 93 | 82.43 |
| PointTransformer V2 [28] | ✗ | 82.8 | 45.15 | 87.52 | 42.73 |
| PointTransformer V3 [29] | ✗ | **87.51** | 61.37 | 94.45 | 74.09 |

Material segmentation experiments are conducted using 2048-point RGB point clouds, classifying materials into one of 13 coarse-grained classes. As shown in Table 4, the models demonstrate robust performance, with PCT and PointTransformer V3 achieving the highest mIoU scores of 98.88% and 98.54%, respectively. This high mIoU across the models indicates that the provided RGB data offers sufficient information for effective material classification, making further segmentation architecture adjustments unnecessary.

Table 4: Models performance on Material segmentation evaluated on 10 Compositions where we report the Accuracy and mIoU

| Model | Fine Grained | | Coarse Grained | |
|---|---|---|---|---|
| | Accuracy (%) | General mIoU (%) | Accuracy (%) | General mIoU (%) |
| PointNet++ [24] | 97.98 | 94.17 | 98.68 | 96.42 |
| PCT [25] | 99.79 | **98.88** | 99.65 | **99.07** |
| CurveNet [26] | 85.07 | 74.26 | 68.75 | 62.04 |
| PointBert [27] | 96.9 | 89.28 | 97.59 | 91.81 |
| PointTransformer V2 [28] | **99.56** | 98.32 | 99.53 | 96.75 |
| PointTransformer V3 [29] | 99.48 | 98.54 | **99.76** | 98.3 |

## 4.3 Grounded Compositional Recognition

The Grounded Compositional Recognition (GCR) benchmark assesses models based on their ability to correctly predict shape categories and part-material pairs, both independently and in a segmented context. The metrics used include Shape Accuracy, reflecting the proportion of correctly predicted shape categories; Value, indicating the accuracy for individual part-material pairs; and Value-ALL, measuring the accuracy of predicting all part-material pairs for a given shape correctly. Additionally, we evaluate the Grounded variants: Value-GRND and Value-ALL-GRND, which consider the accuracy of these predictions when parts are correctly segmented, requiring a part segmentation mask IoU of at least 0.5 with the ground truth.

Analyzing the results in Table 5, PointTransformer V3 achieves the highest shape accuracy of 88.48% and outperforms other models in both fine-grained and coarse-grained tasks, with notable scores in Grounded-value-all at 42.80% (fine-grained) and 72.91% (coarse-grained). PointNet++, particularly with a shape prior, shows robust performance in the coarse-grained task, achieving 58.49% in Grounded-value-all.

PCT, despite its lower shape accuracy, demonstrates effective performance using the shape prior baseline for part segmentation, with results of 26.22% and 60.58% in fine-grained and coarse-grained settings, respectively. CurveNet, while consistent, shows moderate performance, particularly in coarse-grained tasks, and presents areas for potential optimization. These findings highlight the importance of shape priors in enhancing 3D model training and underscore the utility of adapted metrics from compositional recognition frameworks for evaluating complex recognition tasks.

Table 5: **Grounded Compositional Recognition (GCR).** Results of several pipelines on the GCR benchmark, showcasing metrics for shape accuracy Value, Value-All, Value-Grounded, Value-All-Grounded on the Fine and Coarse Grained parts set.

| Semantic level | Model | Prior | Shape Acc. | Value | Value-all | Grounded-value | Grounded-value-all |
|---|---|---|---|---|---|---|---|
| *Fine-grained* | PointNet++ [24] | ✅ | 88.2 | 68.21 | 34.6 | 53.39 | 17.95 |
| | | ⊗ | | 62.02 | 27.15 | 46.42 | 14.09 |
| | PCT [25] | ✅ | 71.59 | 57.92 | 33.96 | 53.77 | 26.22 |
| | | ⊗ | | 43.18 | 17.31 | 37.32 | 11.67 |
| | CurveNet [26] | ✅ | 86.04 | 61.09 | 24.96 | 46.41 | 13.76 |
| | | ⊗ | | 58.29 | 22.43 | 41.87 | 11.23 |
| | PointBert [27] | ✅ | 87.65 | 73.46 | 46.99 | 62.86 | 29.21 |
| | PointTransformerV2 [28] | ⊗ | 85.95 | 70.35 | 40.68 | 60.98 | 26.35 |
| | PointTransformerV3 [29] | ⊗ | **88.48** | **75.72** | **51.65** | **71.84** | **42.80** |
| *Coarse-grained* | PointNet++ [24] | ✅ | 88.2 | 81.41 | 70.75 | 75.1 | 58.49 |
| | | ⊗ | | 55.89 | 32.93 | 42.09 | 19.14 |
| | PCT [25] | ✅ | 71.59 | 68.23 | 63.88 | 66.85 | 60.58 |
| | | ⊗ | | 60.75 | 50.71 | 57.1 | 44.83 |
| | CurveNet [26] | ✅ | 86.04 | 56.17 | 32.47 | 46.86 | 27.32 |
| | | ⊗ | | 54.4 | 30.31 | 44.34 | 24.39 |
| | PointBert [27] | ✅ | 87.65 | 81.88 | 75.74 | 77.63 | 66.80 |
| | PointTransformerV2 [28] | ⊗ | 85.95 | 75.12 | 61.71 | 66.23 | 46.49 |
| | PointTransformerV3 [29] | ⊗ | **88.48** | **82.62** | **77.77** | **80.34** | **72.91** |

## 4.4 Compositional Shape Retrieval

We introduce a new task enabled by our dataset that involves retrieving shapes based on compositional descriptions, using architectures such as ULIP[11], Uni3D[22], and OpenShape[20]. For each shape, metadata from part-material pairs generate template captions, enhanced with detailed color descriptions for each of the 293 fine-grained materials using GPT-4. Captions include descriptions of up to 6 parts to fit the 77-token limit of the OpenCLIP(ViT-G/14) [30] text tokenizer.

In all experiments, we train models using 10 compositions per shape to evaluate retrieval performance across various levels of compositional complexity (1, 3, and 6 parts). Each model encodes both the category text and the shape's 3D point cloud (sampled at 8,196 points and enriched with color information) to create embeddings that capture both shape structure and material details. The retrieval process involves matching these embeddings, with the model predicting by comparing the similarity between the encoded category description and the shape's 3D point cloud.

Table 6 shows shape-to-text classification accuracy in general terms, with Uni3D[22] achieving the highest Top1 and Top5 scores of 62.85% and 86.49%, respectively, demonstrating strong capability in accurately associating complex shape descriptions with corresponding 3D representations.

Table 7 provides detailed results for part-based shape retrieval (R@1 and R@5) in configurations with 1, 3, and 6 parts. These results emphasize the importance of compositional complexity and detailed captions in enhancing model comprehension and retrieval accuracy.

Figure 3 illustrates model performance on the test set after training with 10 compositions. While the model often retrieves objects with correct part-color accuracy, some examples show that it focuses on compositional properties despite misalignments in category, underscoring the dataset's utility in facilitating compositional retrieval and the need for refinement in retrieval methods.

Table 6: Shape classification accuracy metrics (Top1 and Top5) on compositional descriptions. The models predict the correct shape category by matching the encoded 3D point cloud representation with the textual category description.

| Method | Top1 (%) | Top5 (%) |
|---|---|---|
| ULIP[11] | 59.75 | 84.63 |
| Uni3D[22] | **62.85** | **86.49** |
| OpenShape[20] | 61.23 | 84.34 |

Table 7: Detailed part-based shape retrieval performance (R@1 and R@5) across configurations of 1, 3, and 6 parts. Retrieval performance is evaluated by matching the compositional description with the shape's encoded 3D point cloud and color information.

| Method | 1 Part | | 3 Parts | | 6 Parts | |
|---|---|---|---|---|---|---|
| | R@1 (%) | R@5 (%) | R@1 (%) | R@5 (%) | R@1 (%) | R@5 (%) |
| ULIP[11] | 23.04 | 52.36 | 37.41 | 70.27 | 54.8 | 86.6 |
| Uni3D[22] | **26.74** | **58.36** | **39.64** | **74.23** | **59.4** | **88.2** |
| OpenShape[20] | 24.53 | 55.85 | 37.29 | 72.8 | 56.62 | 84.45 |

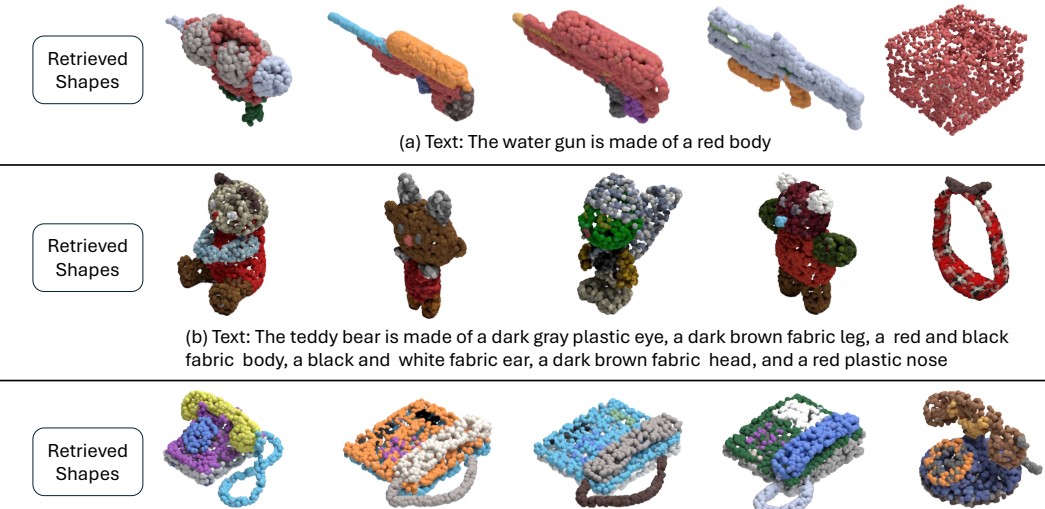

(a) Text: The water gun is made of a red body

(b) Text: The teddy bear is made of a dark gray plastic eye, a dark brown fabric leg, a red and black fabric body, a black and white fabric ear, a dark brown fabric head, and a red plastic nose

(c) The telephone is made of a dark gray metal base, a purple plastic dialing button, a dark speckled brown metal feature button, a bright blue plastic headset, a pale blue plastic headsetcord, and a deep green plastic top.

Figure 3: Qualitative examples for Compositional Shape Retrieval

## 4.5 Computational Resources and Training

The classification and segmentation tasks utilize a single NVIDIA A100 GPU for 250 epochs, with a learning rate starting at $1 \times 10^{-3}$ and decreasing by an order of 0.7 every 20 epochs. The ULIP experiments, more resource-intensive, deploy four A100 GPUs over the same number of epochs, beginning with a learning rate of $5 \times 10^{-4}$, adjusted using a cosine scheduler. The allocation of increased computational resources for the ULIP experiments is justified by the significant improvements in model performance, highlighting the scalability and resource demands of complex text-based retrieval tasks.

**Limitation and Future Works.** While our dataset aims to enhance the granularity of shape categories and parts of 3D shapes to enable 3D compositional understanding, we do not cover all possible categories due to a lack of suitable styling and the possibility of duplicate shapes appearing in the data. In addition, despite the focus of the experiments being the 3D compositional understanding, our dataset can become a valuable asset for compositional 3D shape generation tasks such as Text to 3D and Image to 3D shape.

## 5 Conclusion

In this paper, we present 3DCoMPaT200, a comprehensive dataset designed for part-level 3D understanding, significantly expanding upon previous datasets in terms of object categories, part categories, and material classes. Our dataset enables 3D compositional understanding which we manage to showcase by evaluating 3D only models on the GCR metrics. To leverage the rich annotations and diverse representations in 3DCoMPaT200, we propose a method of describing the

3D shapes using our collected metadata along with colors collected from GPT4 to get the color of each fine-grained material. We show that this new task makes ULIP [11] able to accurately retrieve shapes based on the compositional textual description along with evaluating it using our benchmark compositional part descriptions.

# 6   Acknowledgements

For computing support, this research used the resources of the Supercomputing Laboratory at King Abdullah University of Science & Technology (KAUST). We extend our sincere gratitude to the KAUST HPC Team for their invaluable assistance and support during the course of this research project. We also thank the Amazon Open Data program for providing us with free storage of our large-scale data on their servers, and the Polynine team for their relentless effort in collecting and annotating the data.

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
