# Supplementary of 3DCoMPaT200: Language-Grounded Compositional Understanding of Parts and Materials of 3D Shapes

## 1 3DCoMPaT200 Documentation and Intended Uses

### 1.1 Overview

3DCoMPaT200 consists of 19,051 3D shapes with 1,031 segmented fine-grained parts, 118 coarse-grained parts, 293 fine-grained materials, and 13 coarse-grained materials with a 1000 fine-grained compositions along with compositional captions. 3DCoMPaT200 is designed to facilitate the 3D compositional understanding task as well as introduce a Compositional Shape Retrieval benchmark. This section documents the dataset in accordance with best practices to ensure transparency, reproducibility, and ethical usage.

### 1.2 Data Organizing

Our 3DCoMPaT200 dataset is organized as follows.

```
root/
├── Compat3D_Shapes.zip
├── Shards.zip
├── train/
│   ├── Fine_compositions_json.zip
│   └── Coarse_compositions_json.zip
├── valid/
│   ├── Fine_compositions_json.zip
│   └── Coarse_compositions_json.zip
├── test/
│   ├── Fine_compositions_json.zip
│   └── Coarse_compositions_json.zip
├── Colors.json
├── Parts_fine.json
├── Parts_coarse.json
├── Materials_coarse.json
└── Classes.json
```

Detailed descriptions for each folder or file are given below.

- **root/** - Top-level directory containing all files and subdirectories.
- **Compat3D_Shapes.zip** - Compressed file containing 3D shape models with style place holder for generating compositions.
- **Shards.zip** - Archive of all rendered images for 10 different compositions.
- **train/** - Contains training data sets.
    - **Fine_compositions_json.zip** - Contains 1000 JSON files detailing fine-grained compositions in the training set.

- **Coarse_compositions_json.zip** - Contains 1000 JSON files detailing coarse-grained compositions in the training set.
- **valid/** - Validation data set directory.
    - **Fine_compositions_json.zip** - Contains 1000 JSON files detailing fine-grained compositions in the validation set.
    - **Coarse_compositions_json.zip** - Contains 1000 JSON files detailing coarse-grained compositions in the validation set.
- **test/** - Directory for test data.
    - **Fine_compositions_json.zip** - Contains 1000 JSON files detailing fine-grained compositions in the testing set.
    - **Coarse_compositions_json.zip** - Contains 1000 JSON files detailing coarse-grained compositions in the testing set.
- **Colors.json** - a JSON file containing the colors of the fine grained materials to be used in generating captions per composition.
- **Parts_fine.json** - a JSON file containing a list of the fine-grained part names with the part in the index representing its labels.
- **Parts_coarse.json** - a JSON file containing a list of the coarse-grained part names with the part in the index representing its labels.
- **Materials_coarse.json** - a JSON file containing a list of the coarse-grained material names with the material in the index representing its labels.
- **Classes.json** - a JSON file containing a list of the class category names with the class in the index representing its labels.

## 1.3 Intended Uses

3DCoMPaT200 is intended for use in academic and research settings, specifically for:

- Training and evaluating Compositional 3D alignment and understanding models.
- Advancing the state-of-the-art in GCR Metrics for part and material segmentation and 3D classification.
- Enabling 3D Generation models to generate more fine-grained compositional shapes

## 1.4 Use Cases

- **Academic Research**: 3DCoMPaT200 is perfectly suited for investigating novel algorithms in 3D Part Segmentation, 3D Material Segmentation, 3D Classification, and 3D Generation.
- **Model Evaluation**: This dataset provides a benchmark for evaluating various 3D Understanding methods, particularly those emphasizing compositional capabilities.
- **Educational Purposes**: The dataset, along with its detailed annotations, is valuable for use in academic courses and workshops to teach sophisticated techniques in machine learning and 3D understanding.

## 1.5 Limitations

- **Annotation Bias**: Although human verification is used to maintain high-quality labeling, there may still be biases in the part labeling and the mapping from fine to coarse parts due to subjective human perspectives.

## 1.6 Ethical Considerations

- **Privacy and Sensitivity**: The dataset consists of non-sensitive 3D created and labeled shapes with no connection to specific people or privacy.
- **Use Restrictions**: Users are encouraged to use 3DCoMPaT200 responsibly and ethically, particularly when using it for 3D understanding or 3D Generation.

## 1.7 Documentation and Maintenance

- **Versioning**: A comprehensive version history of the dataset will be kept to monitor changes and enhancements over time.

- **Community Involvement**: User feedback is welcomed to enhance the dataset's quality and suitability for diverse applications.

## 1.8 Statements for NLP

We employ GPT-4V [1] to generate material color descriptions.

## 1.9 Accountability Framework

To promote responsible use and ongoing enhancement, an accountability framework has been implemented. Users of 3DCoMPaT200 are urged to report any issues or biases they discover, thereby contributing to the continuous refinement of the dataset and its annotations.

## 2 Dataset Collection Details

- **Source datasets**: The data is collected through a collaboration with an industry partner where a subset of the shapes was collected from online sources undergoing necessary changes to create proper shapes for our use case and another subset was designed from scratch. We provided a list of guidelines for each category and its associated fine parts segmentation. After the meshes are prepared, we go through an intensive verification of the labels and their suitable material assignation. In addition, we prepare the fine-to-coarse parts mapping and the suitable materials for the coarse-grained parts. each group of similar parts is assigned a common coarse part class that best describes it as shown in figure 1. The code for rendering, and creating 3D compositions is provided at `https://github.com/3DCoMPaT200/3DCoMPaT200/`.

- **Compositional descriptions**: The code for generating captions for shapes in a subset of the compositions is provided at `https://github.com/3DCoMPaT200/3DCoMPaT200/`

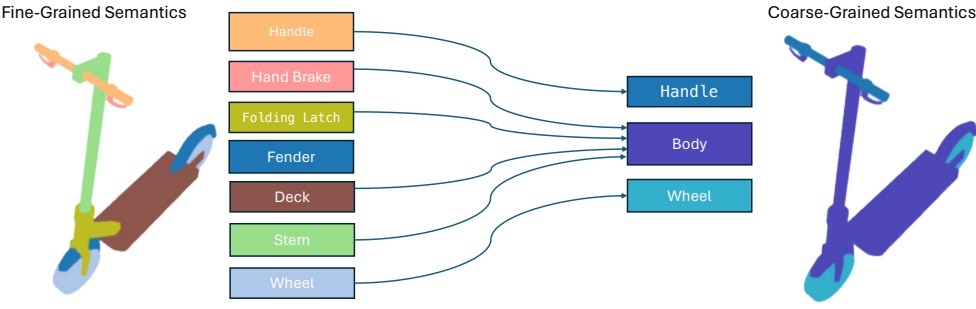

Figure 1: Illustrating Fine-grained to Coarse-grained mapping on a scooter shape

## 3 URL to Data and Metadata

The 3DCoMPaT200 dataset is available for access and download via HuggingFace, offering in-depth views of the dataset components and their annotations. To explore practical examples and download the dataset, visit our Huggingface repository (`https://huggingface.co/datasets/CoMPaT/3DCoMPaT200/`). The dataset's detailed metadata is documented using the Croissant metadata framework, ensuring thorough coverage and adherence to the MLCommons Croissant standards. For metadata specifics, please refer to our Huggingface repository.

# 4    Author Statement and Data License

**Author Responsibility Statement:** The authors bear all responsibilities in case of any violations of rights or ethical concerns regarding the 3DCoMPaT200 dataset.

**Data License Confirmation:** The dataset is available under the [CC-BY-4.0] license, allowing for unrestricted use, distribution, and reproduction in any medium, as long as the original work is properly credited.

# 5    Hosting and Accessibility

The 3DCoMPaT200 dataset is hosted on both GitHub (`https://github.com/3DCoMPaT200/3DCoMPaT200/`) and Huggingface (`https://huggingface.co/datasets/CoMPaT/3DCoMPaT200/`) to ensure consistent and dependable access.

**Maintenance Plan:** The dataset authors will oversee continuous maintenance in case the community reports issues or biases in the labeling for either parts or materials.

**Long-term Preservation:** The dataset is archived on Huggingface (`https://huggingface.co/datasets/CoMPaT/3DCoMPaT200/`) to guarantee long-term availability.

**Structured Metadata:** The annotation for each shape is well documented in a json file for each composition along with the needed part, material, and class labels.

# 6    Data Details

## 6.1    Compositions

The collected parts can be assigned to several suitable coarse materials each of which has several fine-grained sub-material types. This allows leveraging the part-material assignment to create several compositions for each shape. we managed to create 1000 compositions per shape for fine-grained semantic level and 600 compositions for the coarse-grained semantic level. The compositions are illustrated in figure 2.

## 6.2    Rendering

To leverage the meshes compositional value of the dataset as shown in section 4.4. We render 8 views using Blender from different camera parameters: 4 canonical viewpoints and 4 random viewpoints. We first translate each shape above the $z = 0$ plane. In addition, we improve upon the rendering process in 3DComPaT [2] by using BSDF in Blender to make glass reflections for the glass materials as shown in the wine glass example in figure 2.

Each shape is rendered within the same setting, utilizing a single directional light and three area lights arranged around the shape. The shape is positioned on an ovoid surface painted white, ensuring a consistent, uniformly white backdrop. Shadows are cast only on the z = 0 plane where the shape rests. For depth maps and masks, the background surface is omitted from the rendering process. All images are produced at a resolution of 256x256, with 2D images saved in the PNG format. Depth maps are preserved in the OpenEXR format, which supports floating-point representation of absolute distances to the image plane. Segmentation maps for part, coarse material, and fine material labels are first encoded as 11, 5, 7 bits respectively in a 24-bit number with 2 empty bits and then split into 3 8-bit numbers representing the RGB values of the mask to be saved efficiently as PNG file.

## 6.3    Compositional Captions

To enable the Compositional Shape Retrieval, We generate a 10-composition set of captions by leveraging the part-material metadata along with color descriptions obtained from rendering the fine-grained material on a sphere and describing it using GPT4V [1] with examples shown in figure 3. The colors are then merged with the metadata in the format of *The {shape_name} is made of {color} {material_name} {part_name}* as shown in figure 4

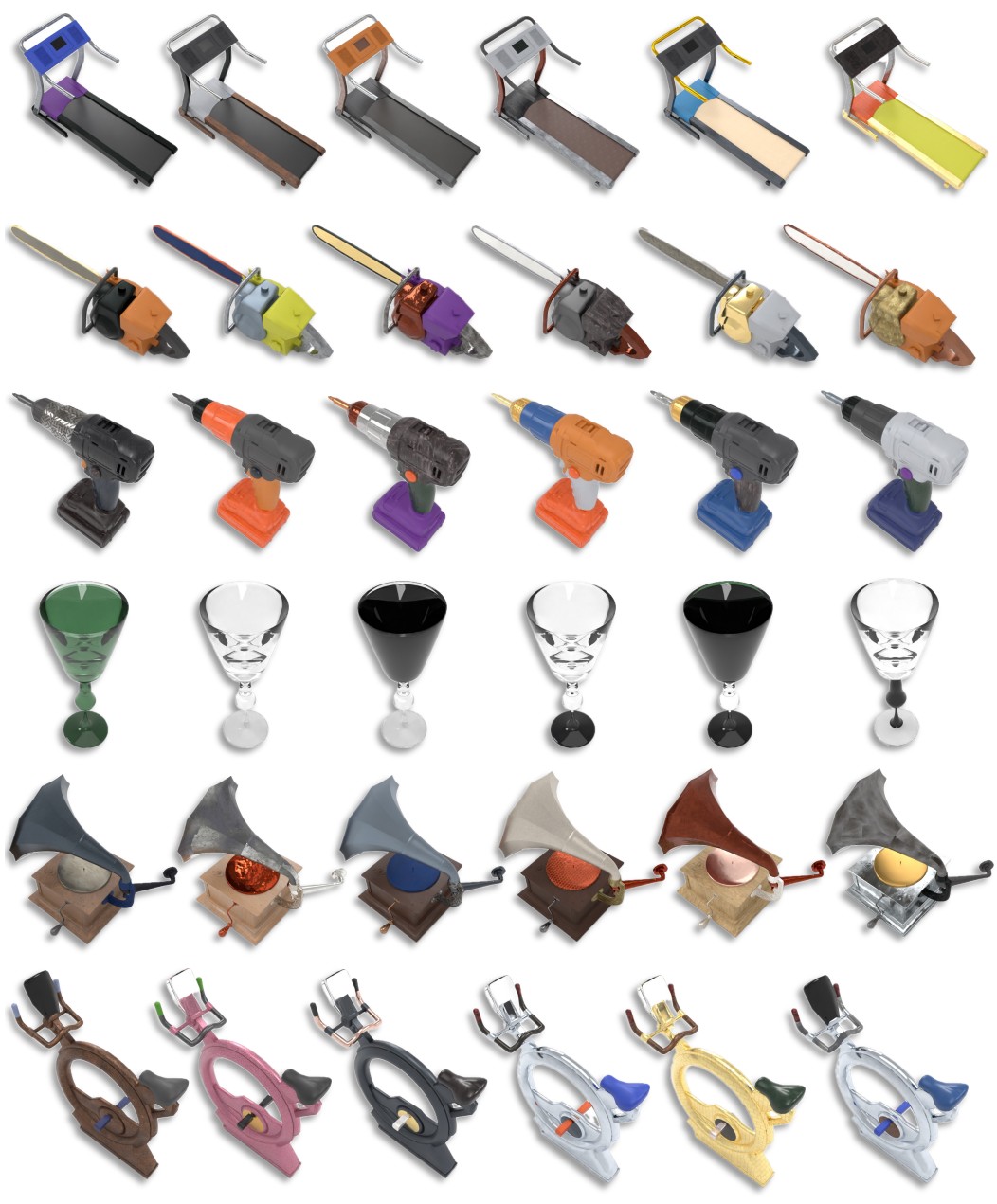

Figure 2: Illustrating 6 compositions for 6 different shapes

For each level of parts in the Compositional Shape Retrieval Benchmark, we create captions with up to 1, 3, and 6 parts per shape to evaluate the R1 and R5 of the model trained on our dataset.

# 7 Qualitative Segmentation results

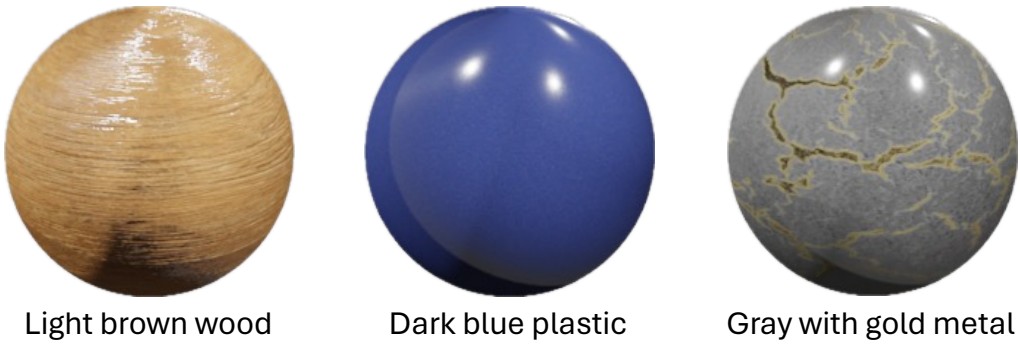

| Light brown wood | Dark blue plastic | Gray with gold metal |

Figure 3: Examples of the rendered materials and their generated GPT4 captions

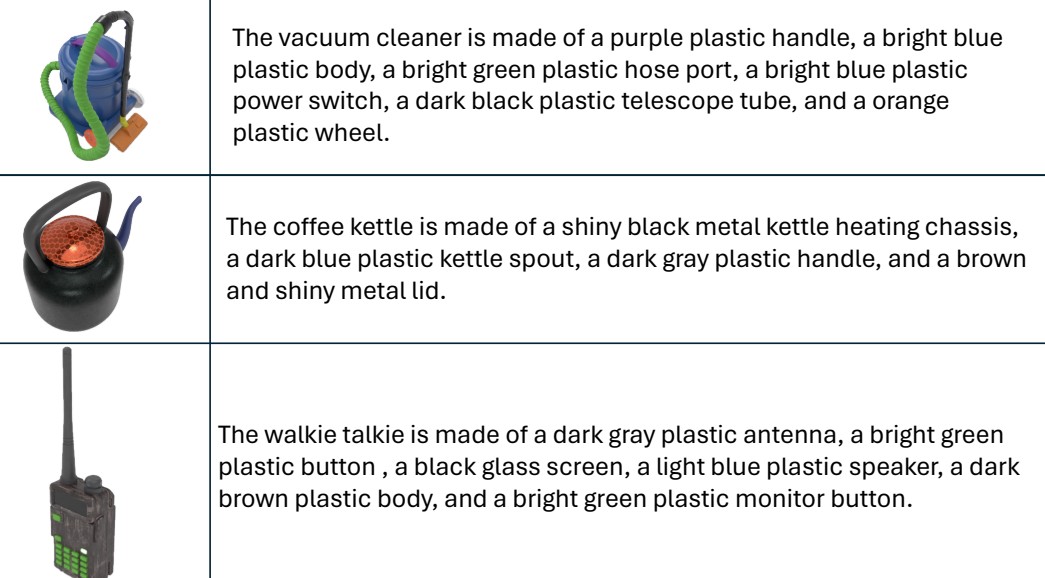

The vacuum cleaner is made of a purple plastic handle, a bright blue plastic body, a bright green plastic hose port, a bright blue plastic power switch, a dark black plastic telescope tube, and a orange plastic wheel.

The coffee kettle is made of a shiny black metal kettle heating chassis, a dark blue plastic kettle spout, a dark gray plastic handle, and a brown and shiny metal lid.

The walkie talkie is made of a dark gray plastic antenna, a bright green plastic button , a black glass screen, a light blue plastic speaker, a dark brown plastic body, and a bright green plastic monitor button.

Figure 4: Examples of the curated captions using the material colors and metadata

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

Figure 5: Qualitative segmentation results on a Tripod shape

Figure 6: Qualitative segmentation results on a Lamp Light shape

| Semantic | Ground Truth | PointNet++ | PointNet++ Shape Prior | PCT | PCT ShapePrior | CurveNet | CurveNet ShapePrior |
|---|---|---|---|---|---|---|---|
| Fine Grained | | | | | | | |
| Coarse Grained | | | | | | | |

Figure 7: Qualitative segmentation results on a Mouse shape