# OpenReview forum: "3DCoMPaT200: Language Grounded Large-Scale 3D Vision Dataset for Compositional Recognition"
_NeurIPS.cc/2024/Datasets_and_Benchmarks_Track — NeurIPS 2024 Track Datasets and Benchmarks Poster_

### Official Review · Reviewer_v45i · 2024-07-18
**Good paper with contributions**

**Rating:** 6
**Confidence:** 3
**Correctness:** Yes. The dataset is constructed in a …
**Clarity:** Yes

**Review:**

The paper is well-written and contributions are presented.

**Strengths:**

1.This dataset encompasses 200 object categories, representing 5 orders of magnitude increase and 1,031 fine-grained parts accounting for 4 times the number of parts compared to its predecessor, 3DCoMPaT.

2.Extensive evaluations demonstrate 3DCoMPaT200’s value in propelling the development of more nuanced and effective 3D object understanding techniques.

3.A captioning dataset is compiled, and a novel compositional shape retrieval task is proposed.

4.Codes for downloading the dataset and conducting the experiments all have been released.

**Additional Feedback:**

No

**Documentation:**

Yes

**Limitations:**

1.Although the proposed dataset additionally collect 160 shape classes and 10 thousand shapes based on 3DCoMPaT. The number of added shapes is relatively small in this LLM and AIGC era.

2.The value of enlarging object categories has not demonstrated clearly. How this dramatically enhance the realism and applicability of simulated environments?

3.The value of the enlarged dataset has not been illustrated clearly. What else community could do based on 3DCoMPaT200 rather than 3DCoMPaT?

**Opportunities For Improvement:**

1.During Section 3.2 Data Collection, the source of CAD shapes used for 3DCoMPaT200 is not illustrated clearly, where only “the same pipeline as the original 3DCoMPaT” is related. Additionally, what about the quality of the CAD shapes compared with Objaverse or Objaverse-XL?

2.Why PCT, PointNet++, and CurveNet are selected for evaluation among so many 3d deep learning networks?

3.As the dataset has 304M rendered images, what’s the quality of these images? Could the rendered images be used for training 2D part segmentation models and utilized for real image part segmentation tasks?

4.As a captioning dataset is compiled, what about the caption’s quality compared with captions generated using Cap3D. Could this dataset be used for text-driven 3D generation and editing tasks, except for compositional shape retrieval tasks?

**Relation To Prior Work:**

Yes

**Summary And Contributions:**

1.This paper presents a large-scale dataset that is designed for compositional 3D understanding of object parts and materials.

2.Extensive evaluations demonstrate 3DCoMPaT200’s value in propelling the development of more nuanced and effective 3D object understanding techniques.

3.A novel Compositional Shape Retrieval task is proposed, which aims to retrieve 3D shapes from compositional part-material descriptions.

---

> ### Author Rebuttal · Authors · 2024-08-17
>
> 1. **The source of CAD shapes used for 3DCoMPaT200 and their quality compared to Objaverse or Objaverse-XL is unclear.**
>
>    The CAD data for 3DCoMPaT200 was meticulously designed and segmented by an industry partner, following comprehensive guidelines tailored to each category. These guidelines included category names, part segmentation instructions, and visual aids to ensure precise annotation. The data undergoes two levels of quality control—first by the industry partner and then by the authors—to verify mesh integrity and correct part labeling. Initial material assignments are also provided by the industry partner, with additional adjustments made as necessary. The PBR materials used are consistent with those in the original 3DCoMPaT. The process of merging fine to coarse-grained parts involves manual inspection and grouping based on name, function, or position, with suitable materials assigned to each coarse-grained part. Comparing our CAD shapes to Objaverse is challenging, as Objaverse includes scanned objects, which inherently have a higher quality compared to designed datasets like ours.
>
> ---
>
> 2. **The rationale for selecting PCT, PointNet++, and CurveNet for evaluation among many 3D deep learning networks.**
>
>    We acknowledge the importance of including diverse models in our evaluation. As such, we have incorporated results for PointBert, MinkowskiResUNet, and PointTransformerV3. Due to the voxelized input format required by MinkowskiResUNet and PointTransformer, their performance currently trails behind methods that directly process point clouds. We plan to further refine hyperparameters to achieve optimal performance across our benchmark dataset.
>
> ---
>
> 3. **Concerns about the quality of the 304M rendered images and their potential use for training 2D part segmentation models.**
>
>    The rendered images were produced using Blender's Cycles engine with 10 samples per render. Each shape was rendered in a custom scene featuring an ovoid white surface, a directional light, and three area lights for balanced illumination. The dataset, provided in 256x256 PNG format, supports 2D part segmentation as demonstrated by our experiments using SegFormer, detailed in Table 5 of the paper. We are exploring the potential for these images to contribute to real-world material segmentation tasks, particularly in enhancing models like Mask2Former and SAM for such applications.
>
> ---
> 4. **Captions Quality and Usage for 3D generation and editing tasks**
>
>    The captions provided in the dataset are formulated using the collected metadata per each shape in terms of the part, and material which are human-annotated unlike the captions provided in Cap3D. the only GPT-generated element in our dataset is the color of each material. Accordingly, the captions can be generated for any number of compositions for our dataset. We believe that this dataset can be useful for material and part editing tasks and we plan to explore the applications of our dataset in these tasks in the future.
>
> ---
>
> 5. **Small size compared to modern datasets**
>
>     We would like to clarify that despite the number of geometric shapes being 19K, the data can generate up to 1000 compositions of styles for these shapes which can be used for 3D shape generation and editing.
>
> ---
>
> 6. **How does this dramatically enhance the realism and applicability of simulated environments?**
>
>     The added categories cover a wide range of objects which are found in most settings and scenes. Using these categories with different styles can help enhance the compositional understanding of 3D LLMs. An application for this can include augmenting Scene datasets such as Scannet which focuses on the same category of shapes as us.
>
>
> Lastly, we extend our gratitude to reviewer v45i for their valuable feedback.

---

### Official Review · Reviewer_VGWS · 2024-07-19
**Good dataset in useful task, but have a key question**

**Rating:** 6
**Confidence:** 2
**Clarity:** Yes, the paper is well-written and ea…

**Review:**

The need for a more extensive dataset of 3D objects annotated with parts and materials is not immediately obvious, but it seems that it could certainly not hurt the progress of methods in the tasks mentioned above (most obviously, 3D shape classification and part segmentation). It also helps greatly that the authors include two additional tasks that can be improved using this dataset. All of these tasks seem to me to be important and relevant tasks in 3D vision today. The paper provides extensive quantitative evaluations that can be used to benchmark future methods, and also provides well thought-out qualitative explanations of the merits of this dataset over previous ones. The dataset and code are publicly available immediately, which helps me verify the validity of this work. Compared to the most famous dataset in this area, ShapeNetPart, this new dataset contains a similar number of total objects (slightly less even?), but with a larger number of object classes. This seems to be both a strength and weakness.

**Strengths:**

In addition the many strengths already listed, it seems that the material annotations of this dataset are the first of its kind and can be very useful for future methods related to that specific area.

**Additional Feedback:**

None.

**Correctness:**

Yes, the claims made in the paper seem to be correct, backed with available data and code.

**Documentation:**

Yes, the data and code are publicly available.

**Ethics:**

No. Ethical concerns are covered in the supplementals, and I agree that there do not seem to be any.

**Limitations:**

Yes, one limitation seems to be mentioned, namely that the dataset could cover more object categories. However, I have a more pressing question, as I have explained above.

**Opportunities For Improvement:**

Most obviously, it seems that given the dataset has the same number of total objects as ShapeNetPart, but with more object classes, there are less objects per class? In 3D vision, there is already the problem of limited data, and it seems that with such few objects per class, the dataset could be dangerously thin and researchers may have to be wary when using this dataset, even *if* all objects in this dataset are of extremely high quality, and well-annotated.

I would like to see a more detailed defense of why this dataset is not too sparse, when some researchers already believe that ShapeNetPart is too sparse to meaningfully advance tasks such as part segmentation due to lack of ability to generalize. This would allow me to potentially raise my rating.

**Relation To Prior Work:**

Yes, the dataset does describe previous datasets, and provides detailed comparisons to them.

**Summary And Contributions:**

This paper presents a large-scale dataset for the task of part-level understanding of 3D objects, as well as understanding of the materials of the objects. The dataset contains 1031 fine-grained part categories and 293 distinct material classes. It builds upon its predecessor, 3DCoMPaT, providing more fine-grained parts and more overall categories. In addition, the paper introduces a brand new task called "Compositional Part Shape Retrieval" to further evaluate and understand the ability of current and future models. This dataset can be thought of as more accurately reflecting the diversity and quality of 3D objects in the real world. One of the main contributions is of course the comprehensive annotations included in the dataset. Quantitative benchmarks are provided in the tasks of: 3D Shape Classification, Part and Material Segmentation, Grounded Compositional Recognition (GCR), and Compositional Shape Retrieval. Finally, the paper includes information about compute, as well as a brief paragraph explaining potential limitations.

---

> ### Author Rebuttal · Authors · 2024-08-17
>
> Thank you for your thoughtful review and for highlighting this concern.
>
> We acknowledge that our dataset features fewer objects per category compared to ShapeNetPart. However, to address this, we offer 1,000 compositions of each shape, which are designed to facilitate a deeper understanding of compositional shapes—an aspect we consider crucial for advancing current 3D language models (LLMs). Our dataset places a strong emphasis on part-level annotations, providing researchers with finer control for category-specific tasks. Additionally, by offering coarse-grained parts, we ensure that the dataset supports a level of understanding comparable to what ShapeNetPart offers, while also enabling the exploration of compositional relationships within the shapes. This approach is intended to mitigate concerns about sparsity and to enhance the dataset's utility for advancing meaningful research in 3D vision and part segmentation.

---

> > ### Comment · Reviewer_VGWS · 2024-08-28
> > **Response**
> >
> > Thank you for your response, I find it appropriate to keep my score of 6 as this is a good work.

---

### Official Review · Reviewer_noGH · 2024-07-24
**Review: 3DCoMPaT200**

**Rating:** 6
**Confidence:** 4
**Correctness:** No concerns regarding the correctness…

**Review:**

In general, this work introduces a high-quality dataset for 3D shapes, which would benefit research in 3D vision and understanding.

Pros (see details below):

1. Clear comparison between the new dataset and prior ones.

2. Fine-grained annotations for parts and materials.

3. Multi-modal integration.

Cons (see details below):

1. Not up-to-date models in evaluation.

2. Unclear motivation about the newly proposed task, Compositional Shape Retrieval.

3. Insufficient visualization or explanation about categories.

**Strengths:**

1. The comparison between the new dataset and prior work is clear, as shown in Table 1.

2. This new dataset provides fine-grained annotations for parts and materials, which could be helpful for enhancing the compositional and semantic understanding of 3D shapes, and further support future research in 3D computer vision and robotics.

3. This new dataset integrates 3D shapes, 2D images, and text descriptions together, enabling multi-modal learning approaches.

**Additional Feedback:**

There are some typos:

- Line 11: “1031” -> “1,031”

- Line 54: This line mentions that there are “1,032 intricate part categories,” but this number is inconsistent with others.

- Line 116: “Shapenet and ObjaVerse” -> “ShapeNet and Objaverse”

**Clarity:**

Yes, this paper is generally well written. There are a few typos (see the additional feedback below).

**Documentation:**

Yes, this work has provided documentation.

**Ethics:**

No ethical concerns.

**Limitations:**

The authors have provided a discussion on limitations and potential societal impacts of this work.

**Opportunities For Improvement:**

1. The experiments for multi-modality alignment are only performed with an early baseline ULIP. It is suggested to include more recent ones, such as OpenShape [Ref1]. [Ref2] and [Ref3] appear more recently (CVPR 2024), so it is understandable not to include experiment results with them, but some discussion could be helpful.

2. Similarly, the experiments for shape classification are performed with early baselines PointNet++, PCT, and CurveNet. More recent and widely adopted models such as [Ref4-6] could be included.

3. The motivation about the newly proposed task, Compositional Shape Retrieval, is not well explained. There are several questions related to the task itself:

	- If the model chooses to retrieve a shape which satisfies the given text description, but is not the original shape that is used to generate the text description, would this retrieval be considered successful?

	- Which practical applications could this task lead to?

	- How does 3DCoMPaT200 compare with prior datasets that also support this new task, if any?

	- What is the “General” accuracy in Table 6? Why is it higher than using all 6 parts of the given shape?

4. It would be helpful to list or showcase the coarse-grained and fine-grained part/material labels, and how the fine labels are associated with the coarse labels in the appendix.

[Ref1] Minghua Liu, Ruoxi Shi, Kaiming Kuang, Yinhao Zhu, Xuanlin Li, Shizhong Han, Hong Cai, Fatih Porikli, Hao Su. OpenShape: Scaling Up 3D Shape Representation Towards Open-World Understanding. In NeurIPS, 2023.

[Ref2] Le Xue, Ning Yu, Shu Zhang, Artemis Panagopoulou, Junnan Li, Roberto Martín-Martín, Jiajun Wu, Caiming Xiong, Ran Xu, Juan Carlos Niebles, Silvio Savarese. ULIP-2: Towards Scalable Multimodal Pre-training for 3D Understanding. In CVPR, 2024.

[Ref3] Zhihao Zhang, Shengcao Cao, Yu-Xiong Wang. TAMM: TriAdapter Multi-Modal Learning for 3D Shape Understanding. In CVPR, 2024.

[Ref4] Xumin Yu, Lulu Tang, Yongming Rao, Tiejun Huang, Jie Zhou, Jiwen Lu. Point-BERT: Pre-training 3D Point Cloud Transformers with Masked Point Modeling. In CVPR, 2022.

[Ref5] Xiaoyang Wu, Yixing Lao, Li Jiang, Xihui Liu, Hengshuang Zhao. Point Transformer V2: Grouped Vector Attention and Partition-based Pooling. In NeurIPS, 2022.

[Ref6] Xiaoyang Wu, Li Jiang, Peng-Shuai Wang, Zhijian Liu, Xihui Liu, Yu Qiao, Wanli Ouyang, Tong He, Hengshuang Zhao. Point Transformer V3: Simpler, Faster, Stronger. In CVPR, 2024.

**Relation To Prior Work:**

Yes, this work clearly describes the key differences.

**Summary And Contributions:**

This work proposes 3DCoMPaT200, a new dataset for 3D shapes with fine-grained annotations for parts and materials. Compared with the prior work 3DCoMPaT, this new dataset includes significantly more object and part categories. Evaluation of several 3D shape models on object classification, part/material segmentation, grounded compositional recognition, and compositional shape retrieval is conducted. 3DCoMPaT200 poses new challenges for 3D shape learning.

---

> ### Author Rebuttal · Authors · 2024-08-17
>
> 1. **The experiments for multi-modality alignment are only performed with an early baseline ULIP.**
>
>     We acknowledge the importance of including more recent methods like OpenShape. We are currently working on fine-tuning the OpenShape pipeline using the PointBert Encoder, though the larger size of the 3D encoder has extended the convergence time. Initial results for 10 data compositions are already presented in Table 5. Additionally, we plan to incorporate a more comprehensive discussion of the latest works from CVPR 2024 in the final version of the paper.
>
> ---
>
> 2. **The experiments for shape classification are performed with early baselines.**
>
>     We appreciate your recommendation and have now included evaluation results for PointBert, MinkowskiResUNet, and PointTransformerV3. Due to the voxelized input format of MinkowskiResUNet and PointTransformer, their performance currently lags behind methods that directly process point clouds. We intend to further optimize hyperparameters to achieve better performance on our dataset benchmark.
>
> ---
>
> 3. **The motivation behind the newly proposed task, Compositional Shape Retrieval, is not well explained.**
>
>     Our benchmark is designed to assess multimodal alignment models based on Recall-1 and Recall-5 scores, specifically using test captions tailored for 1-part, 3-part, and 6-part retrieval scenarios. In accordance with these recall-based metrics, if the retrieved shape does not match the original shape used to generate the description, it will be considered incorrect.
>
> ---
>
> 4. **Which practical applications could this task lead to?**
>
>     This task is aimed at advancing 3D multimodal alignment methods that can better capture compositional details, thereby improving 3D shape understanding. Enhanced 3D-LLM methods, which rely on robust 3D alignment models like ULIP, will benefit from this development.
>
> ---
>
> 5. **How does 3DCoMPaT200 compare with prior datasets that support this new task, if any?**
>
>     The most comparable dataset to ours is PartNet, particularly regarding part segmentation annotations. While some works have added captions to PartNet, it lacks material segmentation metadata for each part. Other datasets, such as Cap3D and the captions provided for Objaverse and ShapeNet by ULIP2, primarily focus on general shape descriptions and do not offer the detailed metadata necessary to support our proposed task.
>
> ---
>
> 6. **What is the “General” accuracy in Table 6, and why is it higher than using all 6 parts of the given shape?**
>
>     "General Accuracy" refers to category-based retrieval, where the model retrieves the shapes whose embeddings best align with the text embedding of a given category. We will update the title of this column in Table 6 to "Category Accuracy." In contrast, the 6-part R@1 and R@5 metrics relate to retrieving a shape based on a detailed 6-part description. Consequently, the results for Category Accuracy are higher than those for the 6-part retrieval task.
>
> ---
>
> 7. **Grammar Mistakes**
>     Thank you for letting us know. We will them in the final version.
>
> Lastly, we extend our gratitude to the reviewer noGH for their valuable feedback.

---

> > ### Comment · Reviewer_noGH · 2024-08-30
> > **Response to Author Rebuttal**
> >
> > The authors’ response is greatly appreciated. The additional evaluation on more recent 3D backbones makes the experiments on 3D shape classification more comprehensive. Two concerns remain:
> >
> > - The “fine-grained” and “coarse-grained” (part and material) labels are not very well-explained in this work. This question is also mentioned by Reviewer YkJk.
> >
> > - Moreover, I agree with Reviewer VGWS on the data sparsity concern. Even though this new dataset features dense annotations for parts, most fine-grained part categories have less than 50 occurrences (Figure 2). It might be too challenging to learn from such a data distribution.

---

### Official Review · Reviewer_YkJk · 2024-07-25
**This paper provide a large and detailed shape part dataset for fine-grained object understanding tasks.**

**Rating:** 7
**Confidence:** 4
**Clarity:** The paper is well written and easy to…

**Review:**

The paper is well-written and easy to follow. The idea of creating a large and detailed part-level shape dataset fosters fine-grained object understanding. The categories of the proposed dataset are 5 times larger than the predecessor with detailed part annotations. The proposed task, Compositional Part Shape Retrieval, provides compositional context for the shape retrieval task.

**Strengths:**

- This paper presents a novel dataset designed for compositional 3D understanding of object parts and materials. It includes abundant object categories and fine-grained part annotations, compared with previous part-level shape datasets.

- Extensive experiments, e.g. object classification, part-material segmentation, and GCR, indicate the proposed dataset would foster the 3D object understanding.

- The dataset also provides compositional descriptions for each object. The authors design a new benchmark, Compositional Shape Retrieval, for evaluating the capability of SOTA methods to understand the structure of the object parts.

**Additional Feedback:**

nil

**Correctness:**

- The claims made in the submission are almost correct. The dataset is constructed soundly, including abundant annotations, e.g. parts and materials.
- The differences between fine-grained and coarse-grained parts need some specific explanations.
- In Tab.4, do these methods use shape prior for the material segmentation? Would shape prior influence the results?

**Documentation:**

The documentation is sufficient and the dataset is available by the given URL.

**Ethics:**

No, there are no or only very minor ethics concerns.

**Limitations:**

The authors have addressed some limitations at the end of section 4.

There are two more questions about the capabilities of the proposed dataset:

- The materials are represented by the color in point clouds. Is the color enough to represent the material? Besides, in the shape retrieval task, the caption includes both color and material information. It seems that the materials are redundant. Please discuss the relationship between color and the materials.

- The captions are generated by the language models, not human annotated. Are there any human descriptions to test the performance?

**Opportunities For Improvement:**

- The captions are generated by the language models. Will the authors add some human descriptions to the dataset?

**Relation To Prior Work:**

This paper discusses the relation to previous part-level annotated shape datasets.

**Summary And Contributions:**

This paper introduces 3DCoMPaT200, a large-scale dataset tailored for compositional understanding of object parts and materials, with 200 object categories, 1031 fine-grained part categories, and 293 distinct material classes. This paper proposes a novel task of Compositional Part Shape Retrieval, using ULIP to provide a strong 3D foundational model for 3D Compositional Understanding.

---

> ### Author Rebuttal · Authors · 2024-08-17
>
> 1. Is the color alone sufficient to represent material in point clouds?
>
>    We assert that color alone in point clouds can sufficiently convey material information in many cases. As evidenced by our results in Tables 2 and 4, utilizing 3D PointClouds improves segmentation outcomes and GCR metrics compared to 2D images, while also offering computational efficiency. However, in shape retrieval tasks, we acknowledge that there may be some overlap between color and material. Nonetheless, relying solely on color is inadequate for capturing the full material properties, as color alone does not distinguish between different materials with similar colors, such as green fabric versus green plastic. The distinct textures and appearances of these materials provide models with a richer and more accurate embedding space.
>
> ---
> 2. Are the captions generated by language models tested against human descriptions?
>
>    The language model-generated captions only specify the color of fine-grained materials. The shape, part, and material names were human-annotated during dataset creation. Each caption follows a template derived from this metadata, with color added by GPT. Regarding the potential redundancy of material information, including it enhances control over shape representation, as identical colors do not necessarily indicate the same material. Different materials, even if similarly colored, possess unique physical properties, such as how they interact with light (e.g., blue metal versus blue plastic), which necessitates distinct handling in the model.
>
> ---
> 3. Can you elaborate on the differences between fine-grained and coarse-grained parts?
>
>    In our supplementary materials (lines 81-84), we outline the grouping process for fine-grained to coarse-grained parts. The primary distinction lies in their level of detail: fine-grained parts contain multiple shape-specific elements suitable for tasks requiring detailed, category-level comprehension, while coarse-grained parts are more generalized to support tasks needing higher-level shape understanding. This is evident in the decrease in category-specific part representation, from 84.19% in fine-grained semantics to 35.59% in coarse-grained semantics. We will ensure this clarification is included in the final paper.
>
> ---
> 4. Does shape prior influence the results in material segmentation?
>
>    The methods referenced in Table 4 of the paper do not utilize shape prior. We found that models converge consistently after 20 epochs, both with and without shape prior, which is why we presented results without incorporating shape prior.
>
> Lastly, we extend our gratitude to the reviewer YkJk for their valuable feedback.

---

### Author Rebuttal · Authors · 2024-08-17

We thank the reviewers and based on their comments we added more experiments for the 3D classification, part segmentation, and material segmentation in Tables 1,2,3. In addition, We also add 2D segmentation results to show that our rendered data can be used for part and material segmentation using SegFormer as shown in Table 5. We also add initial results for fine-tuning OpenShape Checkpoint on our data for 20 epochs. We will be adding the whole comparison between ULIP and OpenShape results in the final paper along with a more detailed discussion of up-to-date methods.

---

### Decision · Program_Chairs · 2024-09-26

**Decision:**

Accept (Poster)

**Comment:**

The authors introduce a new dataset for part-level understanding of objects in 3D named 3DCoMPaT200, with the 200 referring to the new dataset’s 200 object categories 5x larger than the previous 3DCoMPaT dataset. The reviews leaned positive, but somewhat conservatively so, putting this paper near the borderline. Reviewers were generally appreciative of the fine-grained part annotations, especially in category quantity as compared to prior datasets, as well as the material annotations included in the dataset. There were however some common concerns. Notably, the dataset is roughly the same size as prior work in terms of total number objects, but with significantly more categories; this means even fewer objects per category, further exacerbating the data sparsity problem that already exists for 3D datasets. Several reviewers expressed concern about whether current models would be able to effectively learn from such a distribution, given the current state of the field. There were also several concerns about the choice of baselines in the paper, with many more recent models not included; this is quite important for a paper introducing a new dataset in a benchmarks & datasets track. During the rebuttal, the authors provided some initial experiments, but it’s important that these are thoroughly integrated into the paper itself. Overall, the AC leans towards acceptance, but highly encourages the authors to use the feedback from the reviewers to improve the paper where possible.